# AFM Analyses of 3XXX Series Al Alloy Reinforced with Different Hard Nanoparticles Produced in Liquid State

**DOI:** 10.3390/ma13020272

**Published:** 2020-01-08

**Authors:** Verónica Gallegos-Orozco, Audel Santos-Beltrán, Miriam Santos-Beltrán, Ivanovich Estrada-Guel, Iza Ronquillo-Ornelas, José Brito-Chaparro, Ricardo Carbajal-Sanchez

**Affiliations:** 1Departamento de Nanotecnología, Universidad Tecnológica de Chihuahua Sur, Km. 3.5 Carr Chihuahua Aldama, C.P, Aldama 31313, Chih, Mexico; vgallegos@utchsur.edu.mx (V.G.-O.); msantos@utchsur.edu.mx (M.S.-B.); ironquillo@utchsur.edu.mx (I.R.-O.); jbrito@utchsur.edu.mx (J.B.-C.); rcarbajal@utchsur.edu.mx (R.C.-S.); 2Departamento de Metalurgia e Integridad Estructural, Centro de Investigación en Materiales Avanzados (CIMAV), Miguel de Cervantes No. 120, C.P, Chih 31109, Mexico; ivanovich.estrada@cimav.edu.mx

**Keywords:** recycled Al, nanocomposite, stir-casting, nanoparticles dispersion

## Abstract

In the present work, nanocomposites-based 3XXX series Al alloy with three different types of hard nanoparticles, including TiO_2_, C, and CeO_2_, were produced employing two techniques such as mechanical milling and stir-casting method in order to evaluate the viability of integration of the reinforcement in the Al matrix. The integration and dispersion capability of the reinforcement into the Al alloy (3xxx Series) matrix was evaluated, using a phase angle difference and surface roughness analyses by atomic force microscopy operated in both the contact mode (CM-AFM) and tapping mode (TM-AFM), respectively. The distribution profile of both rugosity and the phase angle shift was used to statically quantify the integration and dispersion of the reinforcement into the extruded samples, by using the root mean square (RMS) parameter and phase shift coupled with the events number (EN) parameter. Results from Atomic Force Microscopy (AFM) analyses were corroborated by X-ray diffractometry and scanning electron microscopy (SEM) and transmission electron microscopy (TEM). Microhardness tests were conducted to identify the mechanical properties of the composites in the extruded condition and their correlation with the microstructure. A close relationship was found between the microstructure obtained from the AFM and X-ray diffractometry (XRD) analyses and mechanical properties. Among all, the C reinforcement produced the major changes in the microstructure as well as the best integration and dispersion into the Al-alloy coupled with the best mechanical properties.

## 1. Introduction

The industrial sector has been interested in new materials as composites, particularly the nanoparticulate-reinforced metal matrix composites (nano-MMC’s). These materials offer some key advantages, including: relatively lower production costs, isotropic properties, and the possibility of using a conventional metal forming process such as rolling, extrusion, and forging for manufacturing different components or products [1,2,3]. The aerospace and automotive industries require these composites for key applications considering their combined high strength, low density, and high-temperature resistance characteristics. Nano-MMC’s are materials consisting of a metallic matrix reinforced with hard and insoluble nanoparticles (e.g., oxides, carbides, and nitrides) with a size generally below 100 nm. A significant relevance of these materials is the interaction of dislocations with hard nanoparticles. When added to other strengthening mechanisms as a solid solution, grain size reduction, strain hardening, and precipitation strengthening, which are generally found in nano-MMCs, results in a notable improvement of mechanical properties [4,5,6]. Other authors have reported that, with a small fraction of nano-sized hard particles, nano-MMCs could obtain similar or even much higher mechanical properties than micro-composites [7]. A low concentration of reinforcement also produces low agglomeration at the matrix [8].

The most cost-effective technique of all existing routes for the manufacture of metal matrix compounds is the liquid metallurgy technique, and it is divided into four main categories: pressure infiltration, stir casting, spray deposition, and in situ processing. The stir casting process has several important advantages compared to other routes, e.g., selection of different types of materials (matrix and hard nanoparticles) and a better control of particle-matrix interfacial bonding and matrix structure. The process is simple, low-cost, and flexible and can be applied in large-scale production processes and is suitable in manufacturing processes for near-net shape components [9,10,11,12,13]. Extrusion is a thermomechanical process where severe plastic deformation is produced and is generally used as secondary processing of particulate reinforced composites. This metal forming process promotes the breakup of particle agglomerates and reduce or eliminate the porosity. Each of these help to improve the mechanical properties of nano-MMCs [14].

The aluminum beverage cans are globally one of the most critical recycled products, and the process of recycling takes only 5% of the energy needed to produce new aluminum [15]. In order to take advantage that recycled aluminum can be readily available, this work was focused on knowing the feasibility of integrating different kinds of hard nanoparticles into the Al recycled matrix to increase their mechanical properties. Taken into account a difficulty in integrating nanoparticles into the aluminum matrix during the stir-casting process [16,17], it was necessary to evaluate the microstructural changes produced after the extrusion process and determine the existence of hard nanoparticles into the recycled Al matrix. Nanocomposite-based recycled aluminum (3XXX Series Al alloy) of beverage cans with TiO_2_, C, and CeO_2_ nanoparticles were fabricated using two techniques such as mechanical grinding and stir-casting, using Mg as an auxiliary element in the incorporation and dispersion of the reinforcing phase [18]. This kind of particle shows excellent mechanical properties for MMCs. TiO_2_ is an excellent option because of its good hardness, low density, high melting point, high wear resistance, and good chemical stability [19,20]. On the other hand, the C element, which is transformed into Al_4_C_3_ during the as cast and extrusion process, significantly improve mechanical properties such as yield strength, tensile strength, and Young’s modulus [21]. Lastly, CeO_2_ addition greatly increase the hardness, tensile strength, yield strength, and ductility of composites in the as-cast condition [22,23].

## 2. Materials and Methods

Different types of hard nanoparticles of TiO_2_, C, and CeO_2_ with an average size of about 30 to 50 nm were used as a reinforcing material to increase the recycled Aluminum (3XXX Series Al alloy) matrix strength obtained from beverage cans. Mg powder was used as an auxiliary element the in introduction and dispersion of hard nanoparticles. Using a Spex 8000 mixer/mill in the argon atmosphere, a mixture of 0.5 g of Mg powder, and 6 g of powder in a weight ratio of metallic pure Al/hard nanoparticles of 3:1, were mixed during 5 min and milled using a milling ball to the powder weight ratio of 5:1 for 2 h. The ball mill and vial used were made from hardened steel. The powder mixtures were pressed at a pressure of 10 MPa under uniaxial loading in a hydraulic press during 20 s to get consolidated master compound packets.

An electric resistance furnace equipped with an agitator system of the graphite stirrer impeller was used for melting 200 g of a recycled Al beverage can. The master compound packets composed of pure Al and nanoparticles labeled as Alp-Mg, Alp-TiO_2_, Alp-C, and Alp-CeO_2_, were added in the form of a packet into the molten metal when the vortex was formed every 20 s. The aluminum melted was stirred for 10 min, at the constant rate of 650 rpm, at 800 °C, and poured into cast iron cylinder molds. The final content of Mg and the reinforcement particle in Al-alloy was 0.5 wt.% and 1 wt.%, respectively. Each cylinder was hot extruded by using indirect extrusion at 550 °C where the work metal is forced to flow through a circular die. An extrusion relation of 25 was used under high pressure to produce bars of 10 mm in diameter and 500 mm of length. The samples were polished by sand papers with different grades (Grit 220, 400, 600, 800, 1000, and 1200). Then, for obtaining a mirror finish, samples were polished using 1 µm and 0.5 µm alumina slurry. After that, samples were cleaned with distilled water using an ultrasonic cleaner for 5 min before studying under AFM.

The extruded samples were labeled as Al/Ref, Al/C, Al/CeO_2_, and Al/TiO_2_. Table 1 shows the limit of chemical composition of 3xxx series aluminum alloy that is known from the Aluminum Association. The master powder and extruded composites were studied by X-ray diffraction, Scanning Electron Microscopy (SEM), and Atomic Force Microscopy (AFM). Images of compositional variations were analyzed using WSXM software [24] (WSxM v4.0 Beta 9.3 version, Nanotec Electrónica S.L., Centro Empresarial Euronova 3, Madrid, Spain) recorded from a phase angle difference between the excitation force and the tip response in amplitude modulation of AFM. The diffraction profiles were measured by a Philips X’pert powder diffractometer using a Cu cathode (λ = 0.15406 nm). The step size and step time were 0.02° and 5 s, respectively. The size of the reinforcement from the master compound powder after the milling process together with the crystallite size and micro deformation of the extruded composites were obtained from the analyses of the X-ray diffraction peaks by the Rietveld method. Scanning electron microscopy images were acquired by a cold field emission JEOL JSM-7401 F microscope (JEOL LTD, Akishima, Tokyo, Japan) working at 5 and 17 kV to get images and elemental analysis, respectively. This SEM has an energy dispersive X-ray spectrometer (EDS) facility (Oxford Inca model, Oxford Instruments, High Wycombe, UK). Topography surface characterization was made using an atomic force microscopy tapping mode operated at 10 kHz to 1 MHz of the drive frequency range (Veeco Instruments, Inc., atomic force microscopy, Plainview, NY, USA). Transmission electron microscopy (TEM) characterization was performed using a transmission electron microscope (Philips CM-200 (Philips/FEI Company, Eindhoven, Netherlands) operating at 200 kV) equipped with a DX4 energy dispersive spectrometer (EDS) (EDAX Ltd., Mahwah, NJ, USA).

## 3. Results

### 3.1. Master Compound Samples Powder Characterization

The XRD patterns from master compound samples after 2 h of milling are shown in Figure 1a. The diffractions patterns show the presence of the Al, Mg, and the correspondent phase: C, CeO_2_, and TiO_2_. The diffraction peak width broadening observed is a consequence of the decrease in the size of both crystallite of the Al matrix and reinforcement phase during the milling process. The average size of the reinforcement phase (integrated into the Al matrix of the master compound) was estimated from the Rietveld refinement, which is a detail of the diffraction pattern of the Alp-TiO_2_ sample from Rietveld refinement analysis is shown in Figure 1b. The average size of the reinforcement is shown in Table 2, and the average size of C and CeO_2_ reinforcement phase was around 65 and 40 nm, respectively. The TiO_2_ phase’s average size was ~18 nm.

### 3.2. XRD Characterization of Extruded Samples

During the extrusion process, at higher rates of deformation, the rearrangement of dislocations during the recovery process coupled with the fine-sized reinforcement dispersion, induce the formation of a nanostructured state with a high presence of micro-strains (or high dislocations density) and reduced crystallite size. The reduction in grain size with increased content of reinforcement can be attributed to its distribution and the pinning effect [25,26,27]. X-ray diffraction patterns of composites after the extrusion process at 550 °C are shown in Figure 2. In general, all the patterns only show the Al phase with a preferential orientation of the grain at [111] planes along the extrusion direction. All diffraction peaks of samples show an evident broadening, which was the result of the presence of lattice micro-strains and the presence of reduced crystallite size. The average micro-strains values were evaluated from the X-ray diffraction patterns by using the Rietveld refinement method. From the analyses, we found the best refining results were obtained when only the micro-strains values were considered as the leading cause of the broad diffraction peaks. Therefore, the effect of crystallite size was neglected. Table 3 shows the micro-strains values and lattice parameter for each composite. A correlation between the reinforcement addition and lattice micro-strains values was observed. The samples with reinforcement content showed an evident increase in micro-strain values concerning the Al/Ref sample. Among all the samples, the Al/TiO_2_ sample showed the highest micro-strain lattice value, which suggests the TiO_2_ reinforcement has a strong effect on the aluminum alloy matrix microstructure.

### 3.3. Extruded Samples AFM Topography Characterization

The influence of particle dispersion on surface roughness is shown in AFM topography images and height distribution profiles of surface roughness (see Figure 3, Figure 4, Figure 5 and Figure 6). In general, AFM topography images of samples show the presence of nano-crystallites mostly rounded shaped ranging mainly from ~20 to above 100 nm. Notably, for the Al/Ref sample, the height distribution profile (see Figure 3b) shows a bimodal topographic distribution, which is the product of a non-uniform topography. On the other hand, Al/CeO_2_, Al/TiO_2_, and Al/C composite samples show a more uniform and refined topographic image with a distribution profile located at lower topography values. The height distribution profile of surface roughness was quantified by the RMS (Root Mean Square) roughness parameter, which is the square root of the distribution of surface height topography [28]. According to the results, the presence of the reinforcement produces smaller RMS values (if compared with the Al/Ref sample), which are the result of the refinement topography after the extrusion process. In fact, the RMS values correlate well with micro-strain lattice values found of the characterization of XRD carried out in extruded compounds (see Table 3). The Al/TiO_2_ composite showed the smallest RMS and SD values, which results in uniform dispersion of fine particles of a similar size coupled with the presence of more uniform distribution of small crystallites.

### 3.4. Master Compound Samples AFM Phase Characterization

The integration and dispersion of the reinforcement were statistically quantified from the distribution profile of the phase obtained from AFM phase images. The distribution profile of the phase was first evaluated in the master compound with and without reinforcement content (see Figure 7 and Figure 8), which was followed by the analysis of the extruded samples. The reinforcement phase was located at a phase angle similar to that found in the master compound. Images of compositional variations obtained using WSXM software are shown in Figure 7a and Figure 8a for Alp-Mg and Alp-TiO_2_ master compound samples. Figure 7b and Figure 8b show the degree distribution profile of the phase shift. The continuous line that crosses the clear areas in the phase images coupled with the corresponding distribution profile of the phase change (shown in the inset of Figure 7b and Figure 8b) shows the difference in the phase angle between the Al matrix and another phase. The Alp-Mg master compound sample image (see Figure 7a) shows the presence of clear zones of about 300 nm in size that could correspond to the Mg phase (not found on pure Al), with a distribution profile of a phase shift peak located at ~55° (see the inset of Figure 7b). On the other hand, the Alp-TiO_2_ master compound sample’s phase image shows a topography with uniform, finely rounded grains and the presence of a phase (clear zone) that could correspond to the TiO_2_ phase located at the limit of grains (see Figure 8a). In this case, the peak is located at ~35° of the distribution profile’s phase shift (see the inset of Figure 8b).

### 3.5. Extruded Samples AFM Phase Characterization

The AFM images phase and degree distribution profiles of the phase shift for extruded samples are shown in Figure 9, Figure 10, Figure 11 and Figure 12. The degree distribution profiles of phase shift graphs show the number of events (EN) produced in a specific location by the presence of a second phase. The image phase of the Al/Ref extruded sample (see Figure 9) shows the presence of clear zones between ~10 to ~30 nm. These zones produce a phase distribution profile peak located at around 80°. This is verified by correlating the continuous line that crosses the particle in the phase image (see Figure 9a) and the corresponding distribution profile of the phase shift (inset in Figure 9b). The presence of the Mg phase is related to clear zones, which are similar to the observed in Figure 7 for the Alp-Mg master compound sample image (see Figure 8a,b). For Al/Ref in the extruded sample, these clear zones could correspond mainly to the Al-Mg or MgO precipitates formed during the casting process. The presence of the reinforcement phase in the composites is related to clear zones observable in the phase (see Figure 9, Figure 10, Figure 11 and Figure 12) whose phase distribution peak is located at around 30° (similar to the value observed in Figure 8 for the Alp-TiO_2_ master compound sample). The solid line that crosses some specific clear zones in phase images coupled with the corresponding distribution profile of the phase shift (inset in Figures) shows the type of particle that produces an angle of about 30° (phase angle difference between Al alloy and the reinforcement phase). The event number (EN) of the phase shift produced around 30° of the phase angle difference between the Al alloy and the reinforcement phase, which was used as a quantitative parameter of the dispersion of the reinforcement phase into the Al alloy matrix. The EN and the standard deviation SD of composites are shown in Table 3. The best combination between EN and SD was observed for the Al/TiO_2_ composite, which shows the highest value of EN and the lowest SD. This corresponds to a higher number of fine particles between ~50 to ~80 nm dispersed into the Al alloy matrix and with less presence of agglomerates, respectively (see Figure 11a). On the other hand, the Al/CeO_2_ composites showed the smallest EN value (4700), which means the reinforcing phase is mainly agglomerated. This is manifested in a high standard deviation value (~3.88). Precipitates in the form of fiber seen in Figure 10a could be of the Al-Ce type. These produce a small peak located at around 60° (see Figure 10b). Lastly, the Al/C composite shows the highest EN (17,800) value, which means the presence of the fine second phase disperse into the Al-alloy matrix. This phase produced a distribution peak at around 30° and is related to graphite or the Al_4_C_3_ phase. The other phase, which produces a small signal at around 60° (see Figure 12b), could correspond to the presence of a relatively small amount of Al-Mg or MgO precipitates formed during the casting process, as explained above. This phase corresponds to the bright particles if the size is around 20 nm (see Figure 12a).

### 3.6. SEM Characterization

The SEM images of composites microstructure are shown in Figure 13, Figure 14 and Figure 15. The Al/CeO_2_ composite microstructure (see Figure 13) where the CeO_2_ phase with an average size of ~1 µm is agglomerated and Ce-Al fiber-like intermetallic precipitates are present. Figure 13 also shows a close-up inset view of the CeO_2_ particles. Figure 14 shows the Al/TiO_2_ composite microstructure is observed with a low presence of TiO_2_ agglomerates of ~250 nm and some large precipitates of the Al-Mg-Mn phase. The microstructure of the Al/C composite is shown in Figure 15. The image shows the presence of the Al_4_C_3_ phase precipitates in both a fiber of a few micrometers (~3 µm long) and rounded particles of around 120 nm in size (see the close-up inset in the figure). Fine particles of the Al-Mg phase of about 80 nm were found in the TEM bright-field image into the aluminum matrix of the Al/C composite microstructure (see Figure 16).

### 3.7. Mechanical Properties Characterization

Figure 17 shows the graph of the microhardness value as a function of composite samples and the Al/Ref sample. The composites show an evident increase in the microhardness values concerning the Al/Ref sample. The strengthening of the Al-alloy matrix can be attributed mainly to the combination of both a high micro-strain and fine precipitate dispersion. The Al/C composite shows the highest microhardness value, with an increase of about 50% for the Al/Ref sample. These results are in concordance with a high EN value (~26,000) from AFM phase analyses depicted in Table 3. As we see above, a high EN value indicates the presence of a high size and small size quantity of the second phase. The Al/TiO_2_ composite sample showed an increase of about 28% of microhardness with a marked low dispersion of microhardness values (see Figure 17). The relatively high EN value of ~17,800 and low SD value of ~2.6 (see Table 3) from the AFM analyses is also in concordance with the microhardness results. Small SD values in both microhardness and phase analyses implicate a low presence of agglomerates and precipitates resulting in better dispersion. The AFM image phase of the Al/TiO_2_ composite only shows the presence of one phase (see Figure 11a,b). In the same way, the microstructural analyses SEM revealed a low presence of agglomerates and precipitates of a second phase (see Figure 14). On the other hand, the Al/CeO composite sample only showed an increase of only about 18% with respect to the Al/Ref sample, which could be due to the combined effect of a relatively high value of RMS (~8.69) and low EN (~4700) value. This results in a low microhardness value. The relative high microhardness dispersion data observed in a graph in Figure 17 correlates with the high SD (~3.88) found in AFM phase analyses, which means the presence of agglomerates and intermetallic precipitation observed in SEM analyses (see Figure 13). On the other hand, the Al/Ref sample shows values of micro-strains as relatively low (~0.00014), high RMS values (~12.79 nm), and the EN value at 30° of the phase distribution peak found in his sample was practically zero (see Table 3). This means that the reinforcing phase is not present. The relatively high mechanical properties of the Al/Ref sample could be attributed to the presence of the small precipitates of Al-Mg o MgO phases observed in the phase peak distribution at ~70° of AFM analyses.

## 4. Conclusions

Nanocomposites with a different type of nanoparticles using Recycled Aluminum (3XXX Series Al alloy) from beverage cans has been produced by combining the powder metallurgy and the stir casting method. The microstructure of the extruded samples evaluated by XRD revealed a nanostructured state and the presence of a lattice micro-strain product of the combined effect of the extrusion process and dispersion of the hard particles into Al alloy. The study conducted to estimate the roughness on the surface revealed a roughness surface at a nanoscale level in all the samples. However, for the composites, the presence of the reinforcement further decreased the roughness level displacing the peak of the height distribution profile to the left. The level of the roughness surface was in concordance with lattice micro-strain values found in XRD analyses. The AFM phase analyses revealed the presence of agglomerates and coarse precipitates (quantified by EN and SD parameters), which negatively affected the microhardness values. A close correlation between microhardness and micro-strain lattice values, together with roughness surface analyses, was observed. Among all, the Al–C composite showed the best mechanical properties, which showed better integration and dispersion of the reinforcement coupled with the presence of fine precipitation of Al_4_C_3_. Among different factors affecting the capability of the reinforcement to be homogeneously dispersed throughout the Al matrix during the casting process, is the low wettability of ceramic nano-particles combined with the small size of the reinforcement particle, which tend to form agglomerates. This work proposes a method to evaluate the behavior of the reinforcement in a stir casting process in the nano-MMC’s production.

## Figures and Tables

**Figure 1 materials-13-00272-f001:**
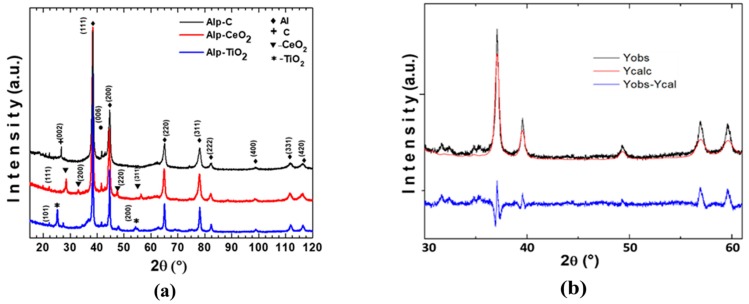
(**a**) XRD patterns from master compound samples after 2 h of milling; (**b**) Rietveld refinement of the Alp-TiO_2_ sample.

**Figure 2 materials-13-00272-f002:**
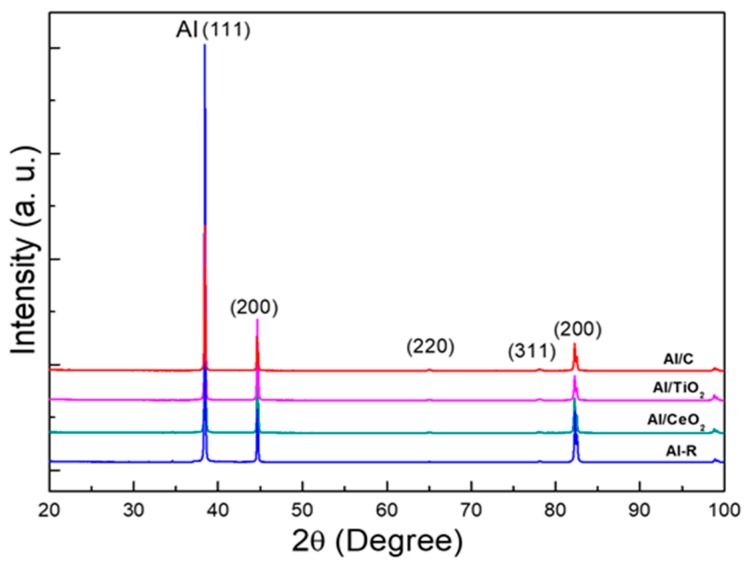
X-ray diffraction patterns of composites after the extrusion process at 550 °C.

**Figure 3 materials-13-00272-f003:**
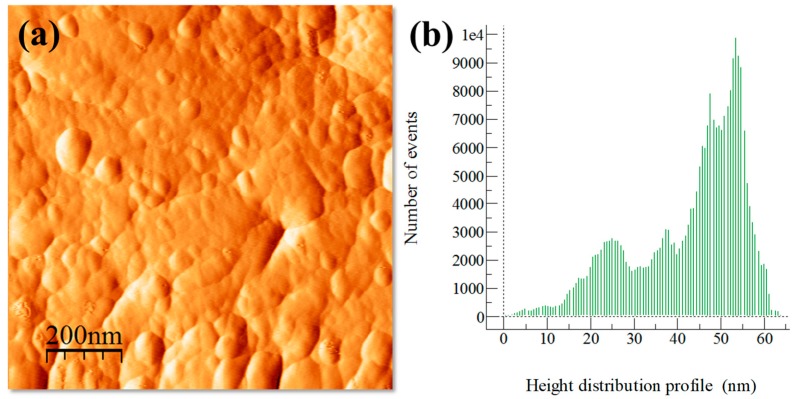
Al/Ref sample after the extrusion process: (**a**) AFM topography image and (**b**) height distribution profile of surface roughness.

**Figure 4 materials-13-00272-f004:**
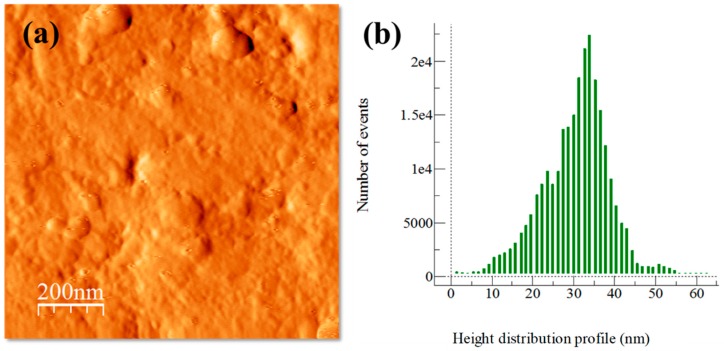
Al/CeO_2_ composite sample after the extrusion process: (**a**) AFM topography image and (**b**) height distribution profile of surface roughness.

**Figure 5 materials-13-00272-f005:**
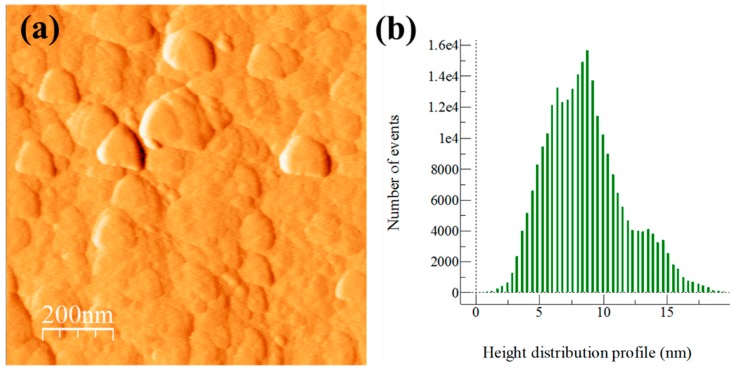
Al/TiO_2_ composite after the extrusion process: (**a**) AFM topography image and (**b**) height distribution profile of surface roughness.

**Figure 6 materials-13-00272-f006:**
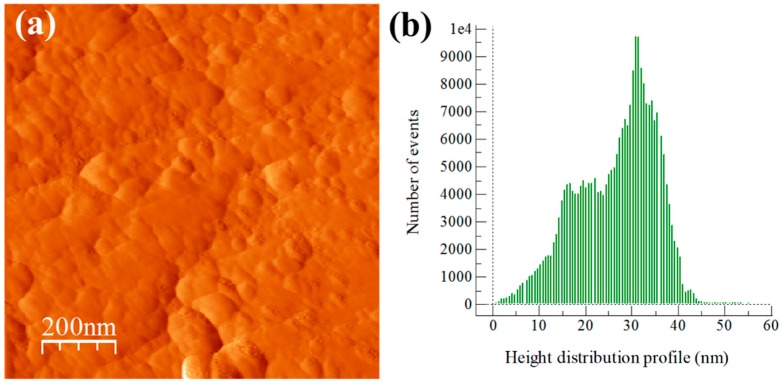
Al/C composite after the extrusion process: (**a**) AFM topography image and (**b**) height distribution profile of surface roughness.

**Figure 7 materials-13-00272-f007:**
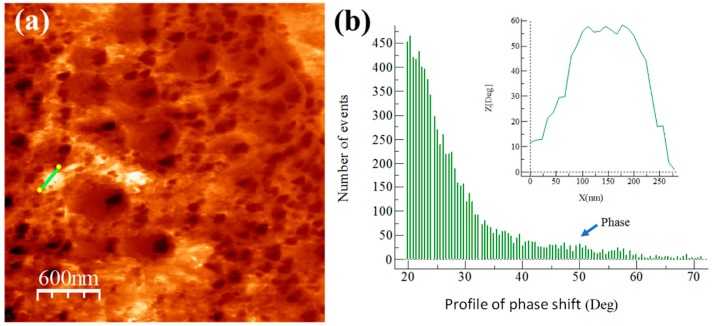
Alp-Mg master compound sample image: (**a**) AFM tapping mode image and (**b**) degree distribution profile of phase shift and distribution profile of a corresponding continuous line that crosses the bright area in the phase image (shown inset of the figure).

**Figure 8 materials-13-00272-f008:**
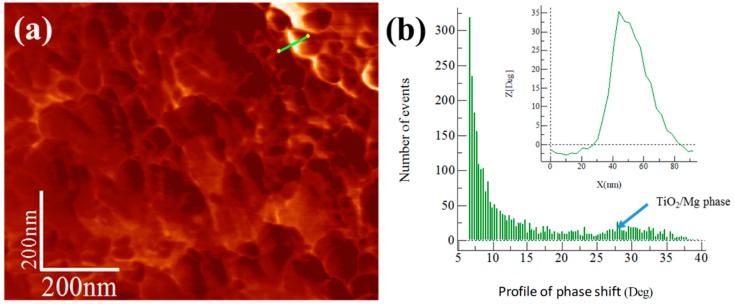
Alp-TiO_2_ master compound: (**a**) AFM tapping mode image and (**b**) degree distribution profile of phase shift and distribution profile of corresponding continuous line that crosses the bright area in the phase image (shown inset of the figure).

**Figure 9 materials-13-00272-f009:**
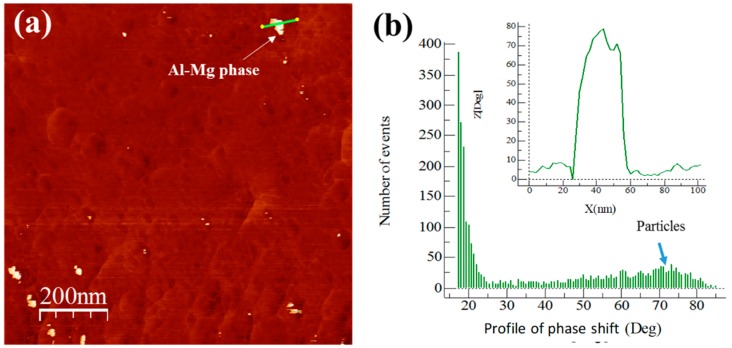
Al/Ref sample: (**a**) AFM tapping mode image and (**b**) degree distribution profile of the phase shift and distribution profile of a corresponding continuous line that crosses the bright area in the phase image (shown inset of the figure).

**Figure 10 materials-13-00272-f010:**
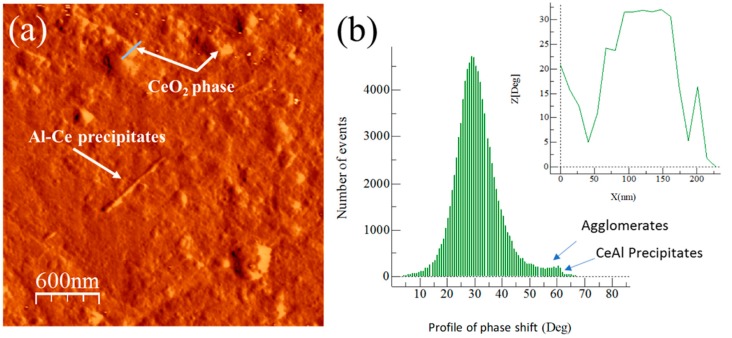
Al/CeO_2_ composite: (**a**) AFM tapping mode image and (**b**) degree distribution profile of the phase shift and distribution profile of a corresponding continuous line that crosses the bright area in the phase image (shown inset of the figure).

**Figure 11 materials-13-00272-f011:**
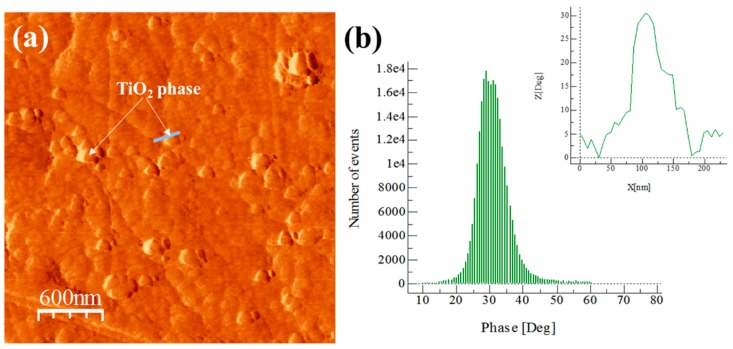
Al/TiO_2_ composite: (**a**) AFM tapping mode image and (**b**) degree distribution profile of phase shift and distribution profile of a corresponding continuous line that crosses the bright area in the phase image (shown inset of the figure).

**Figure 12 materials-13-00272-f012:**
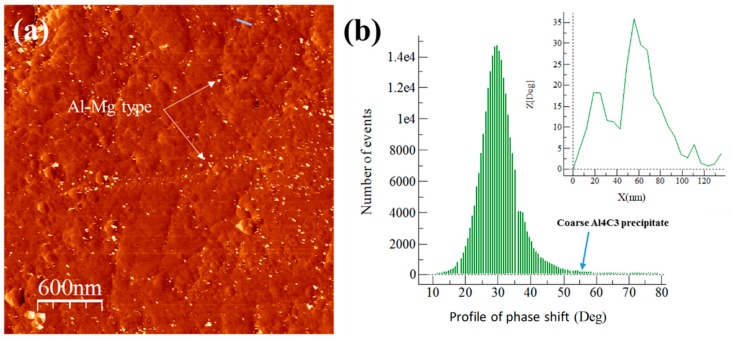
Al/C composite: (**a**) AFM tapping mode image and (**b**) degree distribution profile of the phase shift and distribution profile of a corresponding continuous line that crosses the bright area in the phase image (shown inset of the figure).

**Figure 13 materials-13-00272-f013:**
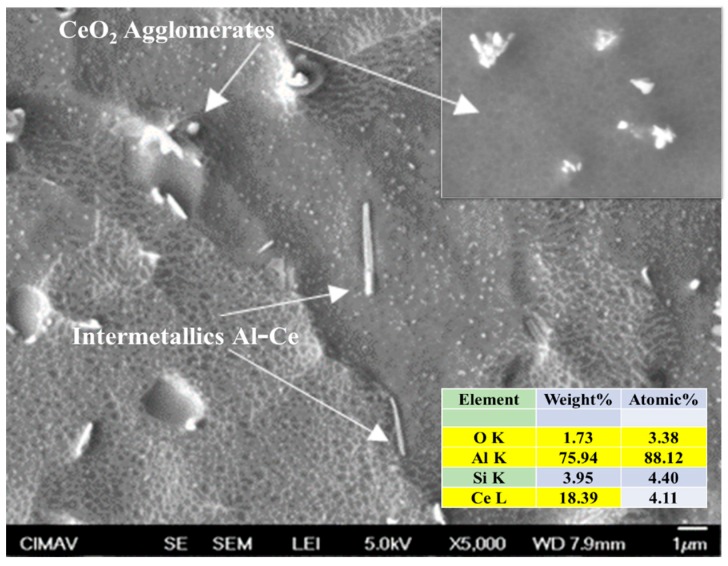
SEM secondary electrons image of the Al/CeO_2_ composite.

**Figure 14 materials-13-00272-f014:**
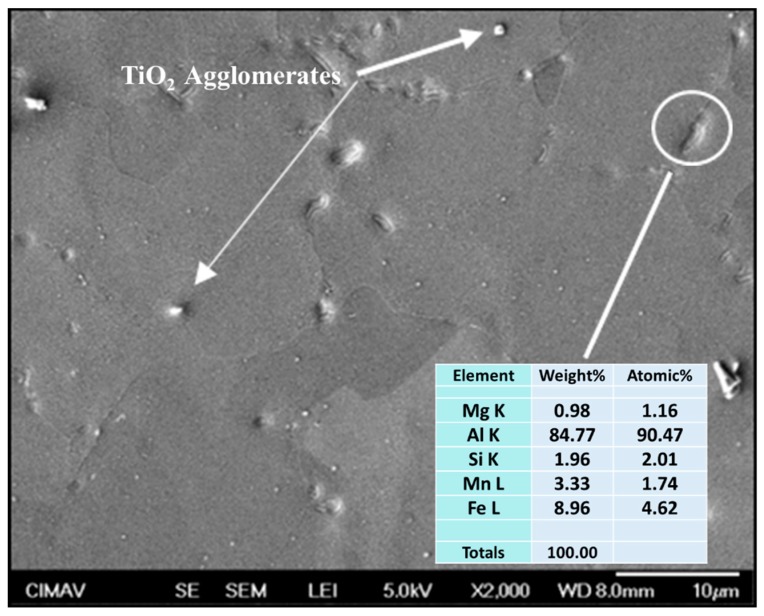
SEM secondary electrons image of the Al/TiO_2_ composite.

**Figure 15 materials-13-00272-f015:**
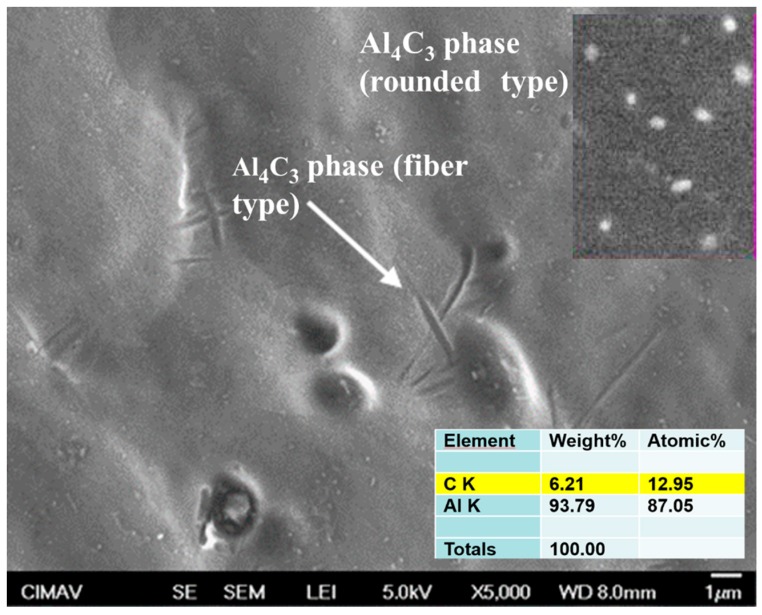
SEM secondary electrons image of the Al/C composite.

**Figure 16 materials-13-00272-f016:**
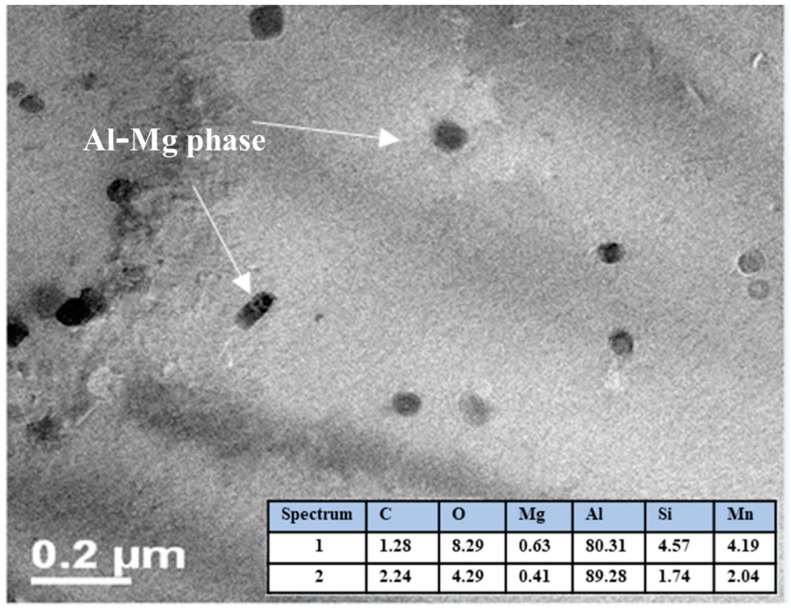
TEM bright field image of the Al/C composite.

**Figure 17 materials-13-00272-f017:**
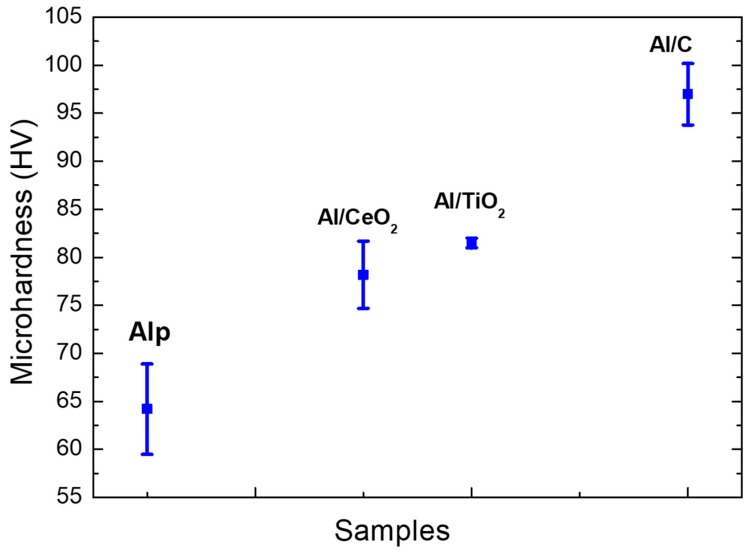
Microhardness values as a function of composite types.

**Table 1 materials-13-00272-t001:** Chemical composition of recycled aluminum.

Weight Percent of Elements						
Al	Mn	Fe	V	Cu	Cr	Zn
Balance	0.74	0.64	0.4	0.13	0.18	0.08

**Table 2 materials-13-00272-t002:** Reinforcement phase size from XRD analyses results.

Phase	Average Size (nm)	Standard Deviation (SD)
C	64.9	0.18
CeO_2_	40.6	0.25
TiO_2_	18.2	0.63

**Table 3 materials-13-00272-t003:** Micro-strain, topography roughness (RMS), and event number (EN) and the correspondent standard deviation (SD) values.

Composites	Micro-strains × 10^−4^ (%)	Topography Roughness RMS (nm)	RMS (SD, nm)	Event Number (EN) (At 30° angle phase)	EN (SD)
Al/Ref	1.4	12.79	±4.88	~0	-
Al/C	5.1	8.54	±3.25	26,000	±3.78
Al/CeO_2_	5.3	8.69	±3.86	4700	±3.88
Al/TiO_2_	8.6	3.10	±1.29	17,800	±2.6

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
