# Peer review of "AFM Analyses of 3XXX Series Al Alloy Reinforced with Different Hard Nanoparticles Produced in Liquid State"

_materials, 2020, doi:10.3390/ma13020272_

Round 1
Reviewer 1 Report
The work concerns the study of composites based on 3XXX Al alloy. Aluminum was obtained from the recycling of beverage cans. The composite was obtained by mixing liquid Al with TiO2, C, CeO2 nanoparticles and the addition of Mg powder as an auxiliary element.
The distribution of nanoparticles, surface roughness after extrusion and microhardness were analyzed.Authors have shown that the combination of classic powder metallurgy and mixing with casting can be sometimes a relatively cheap and effective method of producing nanocomposites.Limited testing of mechanical properties is a certain deficiency. Only microhardness tests do not provide sufficient grounds for inferring the suitability of such composites for industrial use.Performing the stress - deformation characteristics and impact tests would be significantly increase the quality of work.The advantage is to show the structure at the nano level of the composites analyzed, including the density of nanoparticle sizes and their degree of dispersion in the structure.I suggest publishing the article after making the following additions:
Explain on the basis of what premises the nanoparticles proposed in the paper were adopted? Enlarge the font in Fig. 14 On the basis of what premises the composite filling level was adopted at the level of 0.5 wt% and 1 wt%? How do the authors evaluate the technology of producing of tested nanocomposites. The presence of nanopowder agglomerates, the observed heterogeneity of the structure or the visible orientation of the structure in the direction of flow, may indicate of the adopted technology imperfections (mixing method and time, temperature, powder dosing method, etc.)? The authors' opinion on this subject should be included in the conclusions.
Author Response
Explain on the basis of what premises the nanoparticles proposed in the paper were adopted?
Thanks for your observation. In the last paragraph of the introduction section, this paragraph was added:
This kind of particles shows excellent mechanical properties, for example, for MMCs, TiO2 is an excellent option due to its good hardness, low density, high melting point, high wear resistance, and good chemical stability [1, 2]. On the other hand, the C element, which is transformed into Al4C3 during the as cast and extrusion process, significantly improve mechanical properties including yield strength, tensile strength, and Young’s modulus [3]. Finally, CeO2 addition greatly increase the hardness, tensile strength, yield strength and elongation of composites in as-cast condition [4, 5].
Enlarge the font in Fig. 14 (Revisor 1)
Thanks for your observation. The text of the table inside the figure 14, was enlarged.
On the basis of what premises the composite filling level was adopted at the level of 0.5 wt% and 1 wt%?
Thanks for your observation. At the end of the first paragraph of the introduction section, this paragraph was added:
The reduced size of the reinforcement phase down to the nano-scale is such that interaction of particles with dislocations becomes of significant importance and, when added to other strengthening effects typically found in conventional MMCs, results in a remarkable improvement of mechanical properties [6]. It has been reported that with a small fraction of nano-sized reinforcements, MMNCs could obtain comparable or even far superior mechanical properties than MMCs [7]. Low concentration of reinforcement also produces low agglomeration at the matrix [8].
How do the authors evaluate the technology of producing of tested nanocomposites.
Thanks for your observation. At the end of the second paragraph of the introduction section, this paragraph was added:
Amongst the various classical metal-forming procedures, extrusion process is generally used as the secondary processing of discontinuously reinforced composites which can lead to the breakup of particle (or whisker) agglomerates, to the reduction or elimination of porosity, and to improved bonding. All of them contribute to improve the mechanical properties of MMCs.
The presence of nanopowder agglomerates, the observed heterogeneity of the structure or the visible orientation of the structure in the direction of flow, may indicate of the adopted technology imperfections (mixing method and time, temperature, powder dosing method, etc.)? The authors' opinion on this subject should be included in the conclusions.
Thanks for your observation. At the end of the conclusion section, this paragraph was added:
Amongst different factors affecting the capability of the reinforcement to be homogeneously dispersed throughout the Al matrix during the casting process, is the low wettability of ceramic nano-particles combined with the small size of the reinforcement particle which tend to form agglomerates. This work proposes a method to evaluate the behavior of the reinforcement in a stir casting process in the MMCs production.

Reviewer 2 Report
An influence of reinforcing particles on microstructure and microhardness of metal matrix composites based on AA3XXX alloy prepared by mechanical milling and stir-casting method is reported in the manuscript. XRD, AFM, SEM and TEM characterizations were performed on extruded materials and main features were compared with final microhardness measurements. A surface roughness measured by AFM was lower in MMC in comparison with a reference alloy. This effect was related to microstrains as measured by XRD analysis. Best mechanical properties (microhardness) were observed in Al-C MMC.
Although the manuscript is dedicated to a study of modern and highly promising materials, the way the results are described and conclusions are made is rather non standard and to a large extend speculative. Manuscript also contains numerous technical errors. The English used in the manuscript must be definitively improved, some paragraphs could be understood only with a really big effort.
The following text contains several suggestions that might improve the text and soundness of the manuscript:
Technical:
Conditions of extrusion are not provided. Characterization of extrusion matrix are missing.
107 : provide scatter of average size values in Tab. 3.
108-114 explain abbrev. C-M, CeO2-M, TiO2-M in Fig. 1
SEM characterization of master compounds could provide better information about the size or morphology of particles after milling.
158 – Topography roughness was not measured with the accuracy mentioned in the Tab.3. Use rounded values.
190 The inset in Fig.8b is just a copy of the main image
190 while there is i different scale on horizontal and vertical axes in Fig. 8a.
It is not clear on which surface the AFM measurements were performed. How was the surface treated before measurements.
Different phases (AlMg, CeAl) are mentioned in Fig. 9-16, however such a phases do not exist (AlMg), or their precipitation is not too probable (CeAl). Authors probably mean Al-Mg based or Al-Ce based phases.
242 The whole paragraph 3.6. mentions SEM bright -field images, however such a contrast is defined for TEM. Do you mean images in SE or BSE?
286 A solid line connecting values of microhardness in Fig. 17 is rather misleading. There are no transitions between compositions.
293 The presence of microstrains due to the extrusion process is mentioned, however, TEM image (Fig. 17) shows no presence of dislocations.
293 Nanostructured state is mentioned, however SEM and TEM images clearly show that grains in the range of several microns and bigger are present in the material.
Formal:
107 Table 2 in the text ↔ Table II in the caption and also all other tables – use uniform notation
142 Al/Ref is used in the text Al/R in Tab. 3
242 The whole paragraph 3.6. mentions SEM bright -field images, however such a contrast is defined for TEM. Do you mean images in SE or BSE?
254-255 TiO2 use superscript for 2 in Fig. 14 the in Fig. 15 (Al4C3)
251-259 No AlMg phase exists, most probably Al3Mg2 phase is present
Author Response
Reviewer 2
Conditions of extrusion are not provided. Characterization of extrusion matrix are missing.
Answer: The following paragraph was added to section 2 of materials and methods
The sintered material was extruded at 773 K by the indirect extrusion where the pellet is forced to flow through a circular die; an extrusion relation of 25 was used under high pressure to produce bars of 10 mm in diameter and 500 mm of length.
107: provide scatter of average size values in Tab. 3
As a consequence of his suggestion, table 3 was modified and the dispersion of the average sizes is shown more clearly, but in the case of macrostrains values, the dispersion could be considered negligible.
Composites |
Micro-strains x 10-4 (%) |
Topography roughness RMS (nm) |
RMS (SD) |
Event Number (EN) (At 30° angle phase) |
EN (SD) |
Al/Ref |
1.4 |
12.79 |
± 4.88 |
~0 |
- |
Al/C |
5.1 |
8.54 |
± 3.25 |
26000 |
± 3.78 |
Al/CeO2 |
5.3 |
8.69 |
± 3.86 |
4700 |
± 3.88 |
-Al/TiO2 |
8.6 |
3.096 |
± 1.29 |
17800 |
± 2.6 |
108-114 explain abbrev. C-M, CeO2-M, TiO2-M in Fig. 1a
Thank you for your observation, the abbreviations C-M, CeO2-M, TiO2-M in Fig. 1a, have been corrected to Alp-C, Alp-Ce and Alp-TiO2 according to the label proposed in materials and methods section.
SEM characterization of master compounds could provide better information about the size or morphology of particles after milling.
Thank you for your observation. Through AFM phase image analysis, it was possible to establish a qualitative correlation of the dispersion of the reinforcing phase in the Al matrix of the extruded samples, by means preliminary analysis of the AFM phase images of the master compound. The main objective of the MEB analysis of the extruded samples was to characterize the kind of phase present in the extruded samples and their relation with the AFM phase analysis.
158 – Topography roughness was not measured with the accuracy mentioned in the Tab.3. Use rounded values.
Thanks for the observation. The answer given in number 107, could answer this question.
190 The inset in Fig.8b is just a copy of the main image.
Thank you for your observation, Figure 8b was corrected, the wrong image was deleted and the correct graphic was integrated.
190 while there is different scale on horizontal and vertical axes in Fig. 8a.
Thank you for your observation, the scale of Figure 8a was corrected.
It is not clear on which surface the AFM measurements were performed. How was the surface treated before measurements.
Answer: The following paragraph was added to section 2 of materials and methods
The samples were polished by sand papers with different grades (Grit 220, 400, 600, 800, 1000, 1200). Then for obtaining mirror finish samples were polished using 1 mm and 0.5 mm alumina slurry. After that, samples were cleaned with distilled water using an ultrasonic cleaner during 5 min before studying under AFM.
Different phases (AlMg, CeAl) are mentioned in Fig. 9-16, however such a phases do not exist (AlMg), or their precipitation is not too probable (CeAl). Authors probably mean Al-Mg based or Al-Ce based phases.
As a result of its observation, the change of AlMg and CeAl by Al-Mg and Al-Ce was carried out, both in the Figures and in the text, as suggested.
242 The whole paragraph 3.6. mentions SEM bright -field images, however such a contrast is defined for TEM. Do you mean images in SE or BSE?
Thanks for your observation, the text was corrected in paragraph 3.6 together with the legend of Figures 13-15. SEM images where obtained from secondary electrons.
286 A solid line connecting values of microhardness in Fig. 17 is rather misleading. There are no transitions between compositions.
Thanks to your observation, the continuous line connecting the microhardness values ​​in Figure 17 was eliminated.
293 The presence of microstrains due to the extrusion process is mentioned, however, TEM image (Fig. 17) shows no presence of dislocations.
The main objective of the TEM image was to give information about the composition, size and morphology of particles present in the matrix of Al, the analysis of the presence of dislocations or interaction of reinforcing particles with dislocations was not part of this characterization.
293 Nanostructured state is mentioned, however SEM and TEM images clearly show that grains in the range of several microns and bigger are present in the material.
The extruded sample is composed of grains of several microns of size as observed in the SEM image and small subgrains (or crystallites) of size, most of them, in the range of ~ 100 to ~ 300 nm (visible in Fig. 13 in the SEM image). These results correspond well with AFM measurements where is expressed that the crystallite size is from ~ 20 to above 100 nm. On the other hand, the topography roughness measurements show values under 100 nm observed in figure 3 to 6.
107 Table 2 in the text ↔ Table II in the caption and also all other tables – use uniform notation.
Table II was modified by Table 2 in both the text and the Table, thanks for your observation.
142 Al/Ref is used in the text Al/R in Tab. 3
Thanks for your observation, it was standardized using the abbreviation Al/Ref, in the Table 3 as in the text.

Round 2
Reviewer 2 Report
English must be improved in the whole manuscript.
60-61 This work is focused on knowing the feasibility of integrating and dispersion hard particles into the Al
recycled matrix to increase their mechanical properties. --> This
work is focused on a verification of a feasibility to integrate and disperse hard particles into Al recycled materials in order to improve their mechanical properties.
68, 70 check for subscripts in the text.
75-78 Reformulate the sentence
89 do not mix °C with K in one sentence.
167-168 except for TiO2 reinforcement, all other values of RMS are the same within the experimental error.
178 Tab. 3: generally, standard deviations are given with two significant digits (in maximum) and results should be then rounded.
228 numbers on axes in the inset of Fig. 7b could not be read. The same 9b and 10b.
325 microns missing in the pdf file
371 TEM bright field images image of the Al/C composite. --> TEM bright field image of the Al/C composite.
References: Read carefully all corrections and remove mistyping errors.
